# Learning and diSentangling patient static information from time-series Electronic hEalth Records (STEER)

**Wei Liao, Joel Voldman***

Department of Electrical Engineering and Computer Science, Massachusetts Institute of Technology, Cambridge, Massachusetts, United States of America

* voldman@mit.edu

## Abstract

Recent work in machine learning for healthcare has raised concerns about patient privacy and algorithmic fairness. Previous work has shown that self-reported race can be predicted from medical data that does not explicitly contain racial information. However, the extent of data identification is unknown, and we lack ways to develop models whose outcomes are minimally affected by such information. Here we systematically investigated the ability of time-series electronic health record data to predict patient static information. We found that not only the raw time-series data, but also learned representations from machine learning models, can be trained to predict a variety of static information with area under the receiver operating characteristic curve as high as 0.851 for biological sex, 0.869 for binarized age and 0.810 for self-reported race. Such high predictive performance can be extended to various comorbidity factors and exists even when the model was trained for different tasks, using different cohorts, using different model architectures and databases. Given the privacy and fairness concerns these findings pose, we develop a variational autoencoder-based approach that learns a structured latent space to disentangle patient-sensitive attributes from time-series data. Our work thoroughly investigates the ability of machine learning models to encode patient static information from time-series electronic health records and introduces a general approach to protect patient-sensitive information for downstream tasks.

## Author summary

It is increasingly apparent that machine learning for healthcare models can predict sensitive information from data that does not explicitly encode it. Well-known examples include self-reported race from various medical imaging modalities, and age and biological sex from retinal fundus images. These findings in turn raise concerns about introducing biases in models or exacerbating health disparities. However, we lack a clear understanding of the extent of the problem—what types of sensitive information can be predicted, how does it generalize to different models or different datasets—and, critically, approaches to develop models that can make clinical inferences but *not* infer sensitive

**Data Availability Statement:** The dataset analyzed in this study can be found in https://physionet.org/content/mimiciv/2.2/ and https://eicu-crd.mit.edu/about/eicu/. The code for this study is available in

https://github.com/weiliao97/Learning_Time_Series_EHR.

**Funding:** This work was supported by MIT EWSC Fellowship (awarded to WL). The funding source had no role in study design, data collection and analysis, decision to publish, or preparation of the manuscript.

**Competing interests:** The authors have declared that no competing interests exist.

information. Here we go beyond these prior studies and thoroughly investigate the ability of machine learning (ML) models to encode a wide range of patient sensitive information from time-series EHR data, and then, critically, provide a strategy to mitigate such inferences.

## 1. Introduction

There is growing adoption of electronic health records (EHR) in the machine learning for healthcare (MLHC) field to provide a more complete picture of patient health, inform treatment decisions, and improve quality of care. EHR contains a vast amount of data, including medical histories, diagnoses, treatments, and outcomes. This data is often collected over time, creating time-series data that can be used to track patient progress and devise effective treatment plans [1–3].

In recent years, there has been emerging evidence reporting biases in MLHC models, highlighting the potential risks that the use of machine learning (ML) could amplify health disparities [4,5], and the importance of identifying and accounting for potential biases in the data [6]. Researchers have also found that from either multiple imaging modalities (X-ray imaging, CT chest imaging, mammography)[7], clinical notes [8] or a combination of vital signs [9], ML models can be trained to predict self-reported race with high area under the receiver operating characteristic curve (AUC). Besides race, studies have found that age and biological sex can be accurately predicted from either retinal fundus images [10,11] or chest X-ray images [12]. These studies highlight the possibility that 1) such patient-sensitive information could be misused, thus undermining patient privacy, and 2) MLHC models may amplify health disparities [13,14] by utilizing this information for downstream decision-making.

Notably, it is still unclear the range of such patient-sensitive information that can be predicted from deidentified medical data. For EHR data, a previous study focused on race [9], leaving open the question of whether these findings could be extended to sex, age and comorbidity factors. Further, questions of generalizability remain unaddressed, such as whether this issue is specific to a particular dataset or whether it is sensitive to different modeling setup. Finally, and critically, there is a lack of algorithmic approaches to mitigate the harm that such data identification could present to patient privacy and model fairness.

In this work, we first investigated sex, age, race and 15 comorbidity factors (which we denote as static data) and found that they can all be predicted from time-series EHR data. Such information could exist in the original time-series variables or even in hidden representations from models trained without access to static information, with AUCs well above 0.8. This finding was not specific to a single task; we observe high predictive performance for models developed to predict either 48h in-hospital mortality or hourly SOFA score. Additionally, we observe that such findings are valid across model architectures, different cohorts and different EHR databases, suggesting an endemic issue with MLHC models utilizing EHR data.

The ability of time-series data to encode patient sensitive attributes poses risks of re-identification of patients in rare subsets and unfair decisions for downstream tasks [15–17]. Therefore, inspired by the approaches developed in the disentangled representation learning field [18,19], we further designed a variational autoencoder (VAE) model to diSentangle patient static information from Time-series Electronic hEalth Records (STEER). VAE provides a powerful framework for learning representations that capture the underlying structure and sources of variation in the data [20–22]. A wealth of research has utilized VAE to learn representations that satisfy certain fairness metrics, either through obfuscating sensitive information [23,24],

training adversarial classifiers [25–27], or enforcing distribution or distance criteria [28–30], all of which were validated on imaging and tabular data. In the metric learning field, this disentanglement idea has also been developed to learn target class image embeddings that are decorrelated from sensitive attributes [31]. However, learning structured representations from time-series data is still unresolved. Our approach with STEER fills this gap and preserves important merits including easy adaptation to multiple sensitive attributes and their conjunctions, and not requiring the sensitive attributes for inference [18]. The latent space in STEER is composed of two subspaces, sensitive latents and non-sensitive latents. We demonstrate that our method can achieve both: 1. disentanglement (the ability of using sensitive latents to predict sensitive labels); and 2. utility (the ability to perform clinical prediction tasks using only non-sensitive latents). The approach is flexible, in that it can adapt to both point prediction such as in-hospital mortality or time-series prediction such as clinical scores. The sensitive latents can be discarded or noised out when performing downstream tasks in order to protect the sensitive attributes from being utilized.

## 2. Results

### 2.1 Predicting static information from time-series EHR

To understand the extent to which static information is encoded in time-series EHR, we first trained models to perform two separate predictions solely using time series data from the MIMIC-IV dataset. We chose distinct outcomes of interest to the clinical community, namely 48h in-hospital mortality (IHM) and Sequential Organ Failure Assessment (SOFA). SOFA score is commonly used to track a patient's status during their ICU stay to determine the extent of a patient's organ function or rate of failure [32]. It creates a standardized, numeric score (0–24) that is familiar to critical care physicians. 48h IHM aims to identify patients who are at high risk of dying during their hospital stay, whereas SOFA score prediction is a time-series task that aims to forecast the severity of organ dysfunction in critically ill patients over time. While hospital mortality prediction focuses on the binary outcome of life or death, SOFA score prediction takes into account the dynamic nature of organ dysfunction in critically ill patients. Both tasks have important implications for clinical decision-making and patient outcomes, but they require different modeling techniques and evaluation metrics. Most importantly, the features learned by models should be relevant to the corresponding prediction target.

For the hospital mortality task, we used the General cohort and for the SOFA prediction task, we used both the General and the Sepsis-3 cohort. Demographics and ICU stay information for both cohorts are in Table 1. For each prediction task, we used 3 different model architectures to ensure results are not specific to a particular architecture. From these models, we observe AUCs for the IHM prediction are between 0.879 to 0.890, which is comparable to prior IHM prediction works using MIMIC-III data [33,34]. Similarly, for the SOFA score prediction task, the root mean square errors (RMSE) are 1.45–1.48 for the General cohort and 1.69–1.72 for the Sepsis-3 cohort, which are reasonable. RMSE in Sepsis-3 cohort is 16.6% higher than the General cohort, mainly due to the average SOFA difference in these 2 cohorts: the Sepsis-3 cohort average SOFA is 20.8% higher than in the General cohort. Clinical guidelines typically use a range of ~2 in SOFA score in their recommendations, which forms the basis for our assessment that a SOFA score MSE between 1 and 2 is reasonable. These results establish a suite of baseline models that we can examine for their ability to predict static information.

Models trained in Table 2 only have access to time-series EHR during training. To determine whether the representations learned through the clinical task prediction can be used to

**Table 1. MIMIC-IV cohort overview.**

|  |  | General<br>n = 38,766 | Sepsis-3<br>n = 18,327 |
|---|---|---|---|
| **Sex** | Female | 16,825 (43.4%) | 7,676 (41.9%) |
|  | Male | 21,941 (56.6%) | 10,651 (58.1%) |
| **Age** | <30 | 1,753 (4.5%) | 849 (4.6%) |
|  | 31–50 | 5,264 (13.6%) | 2,190 (11.9%) |
|  | 51–70 | 15,748 (40.6%) | 6,803 (37.1%) |
|  | >70 | 16,001 (41.3%) | 8,485 (46.3%) |
|  |  | mean: 65.1, std: 16.8 | mean: 66.4, std:16.2 |
| **Race** | American Indian/Alaska Native | 63 (0.2%) | 31 (0.2%) |
|  | Asian | 1,137 (2.9%) | 532 (2.9%) |
|  | Hispanic | 1,319 (3.4%) | 581 (3.2%) |
|  | Black/African American | 3,384 (8.7%) | 1,428 (7.8%) |
|  | White | 26,271 (67.8%) | 12,469 (68.0%) |
|  | Other/Unknown | 6,592 (17.0%) | 3,286 (17.9%) |
| **Hospital Mortality** | No | 35,214 (90.8%) | 15,821 (86.3%) |
|  | Yes | 3,552 (9.2%) | 2,506 (13.7%) |
| **SOFA** | 0–4: | 56.40% | 43.31% |
|  | 4–8: | 32.50% | 39.96% |
|  | 8–12: | 8.64% | 12.69% |
|  | 12–16: | 2.00% | 3.28% |
|  | 16–20: | 0.42% | 0.69% |
|  | 20–24: | 0.04% | 0.06% |

predict static information, we use the activations of hidden layers in these pretrained models and build new models to infer the static information. The approach is also shown in Fig 1A and 1C. Fig 1A demonstrates the hidden representation extraction from TCN model. For the LSTM model, the outputs from the last LSTM layer are extracted. In the Transformer model, the outputs from the transformer encoder are used as inputs for the downstream prediction tasks. Importantly, as a control, we also built another model to predict the same series of target attributes from the original time-series EHR data (Fig 1D).

To understand the range of static variables that can be predicted using models trained solely on time-series data, we attempted to infer 18 different static variables and evaluated the AUC results. 7 out of the 18 test set AUCs are shown in Table 3. Importantly, the test set remains the same in both stages: the IHM and SOFA prediction model evaluation in Table 2 and the static information prediction stage conducted here. As expected, in each condition, predicting from the original time-series has the highest AUC. For instance, in the sex prediction, the original time-series has an AUC 0.857, while the hidden representations from the rest conditions have

**Table 2. Model performances.**

| Task | Cohort | Models | Results (Average, 95% confidence interval, CI) |
|---|---|---|---|
| **48h IHM** | **General** | LSTM | 0.881 (0.860–0.898) |
|  |  | TCN | 0.879 (0.861–0.896) |
|  |  | Transformer | 0.890 (0.873–0.906) |
| **SOFA score** | **General** | LSTM | 1.47 (1.42–1.52) |
|  |  | TCN | 1.45 (1.41–1.49) |
|  |  | Transformer | 1.48 (1.43–1.52) |
| **SOFA score** | **Sepsis-3** | LSTM | 1.72 (1.68–1.77) |
|  |  | TCN | 1.69 (1.65–1.74) |
|  |  | Transformer | 1.71 (1.67–1.76) |

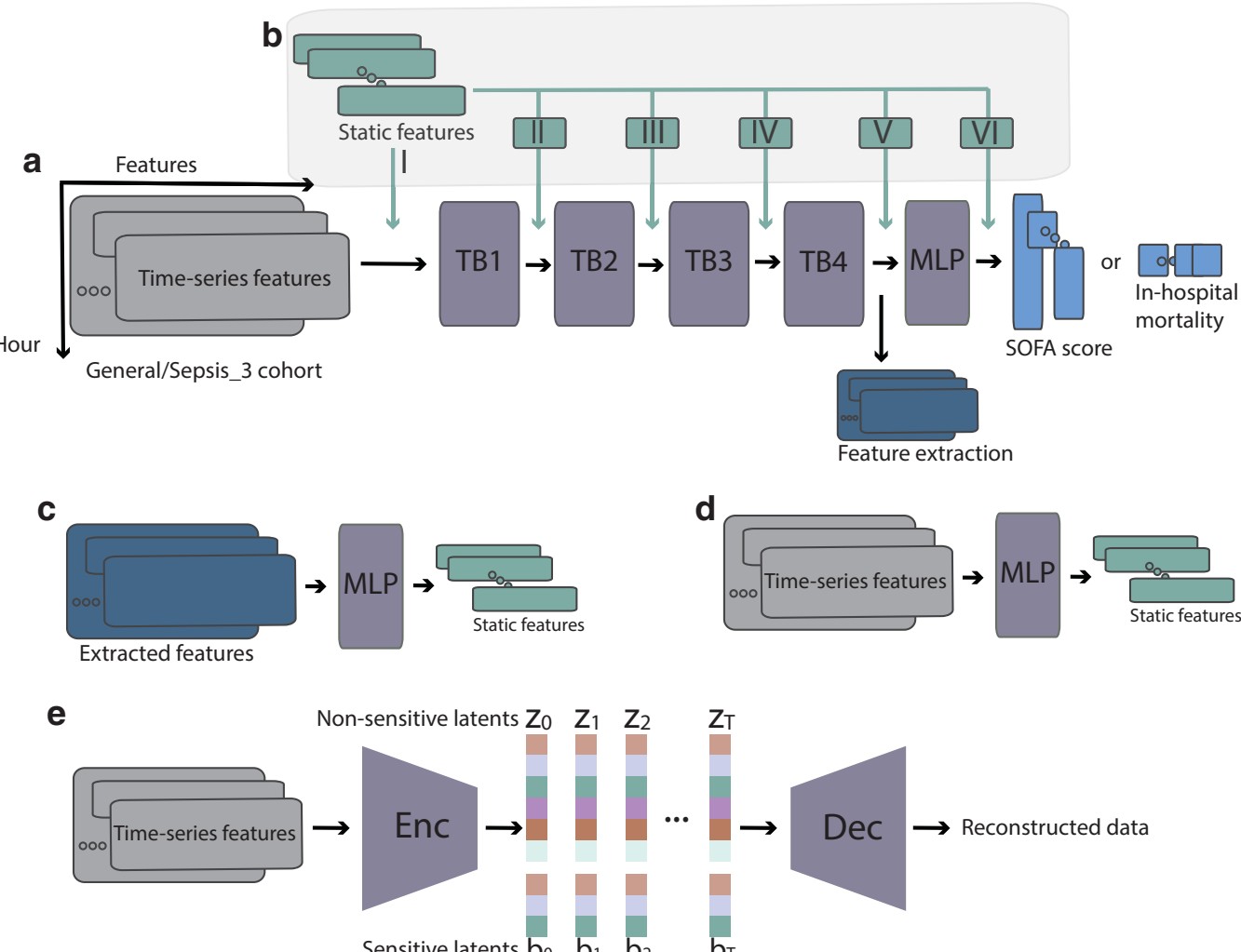

**Fig 1.** a) Time-series data was used to develop models either to predict patient IHM or SOFA score. TCN model architecture is shown as an example. TB: temporal block. MLP: multilayer perceptron. b) Static information was fused at different stages to explore whether it improves SOFA score prediction performance. c) Hidden representations were extracted under different modeling configurations and used to predict static features. d) As a control, the original EHR time-series data was also used to infer static information. e) The VAE model developed to learn a latent space which achieves both disentanglement and utility.

AUCs ranging from 0.823 to 0.851. Among the 7 target static features we showed in Table 3, the AUC performances are all high (0.765 to 0.946) despite the differences in the original model setup (different tasks, different cohorts, and different model architectures). For the entire set of static information prediction performance, please refer to S1 Table–S10 Table. These results demonstrate that a broad set of static information is encoded in time-series data and can be learned by ML models.

Next, we employed data fusion as a complementary method to assess whether the static information is already encoded in the dynamic information. We tested whether fusing these 18 static variables would decrease the SOFA score prediction error. Using the TCN model with Sepsis-3 cohort, we compared different fusion strategies where we fused static information into the main model at different stages (Fig 1B). We found that fusion leads to no decrease in RMSE (Table 4), further supporting the notion that static information is already encoded in the time-series EHR data.

**Table 3. MIMIV-IV test results.** Average and 95% CI reported.

| | | Sex | Age | Race | CHF[1] | Diabetes[2] | Renal[3] | SLD[4] |
|---|---|---|---|---|---|---|---|---|
| Original | | 0.857 (0.854–0.860) | 0.876 (0.873–0.879) | 0.833 (0.828–0.838) | 0.833 (0.828–0.837) | 0.831 (0.827–0.835) | 0.923 (0.921–0.926) | 0.946 (0.941–0.951) |
| **48h IHM** | **LSTM** | 0.832 | 0.860 | 0.799 | 0.801 | 0.809 | 0.907 | 0.932 |
| | | (0.828–0.835) | (0.856–0.863) | (0.794–0.805) | (0.797–0.805) | (0.804–0.813) | (0.904–0.910) | (0.926–0.938) |
| | **TCN** | 0.827 | 0.856 | 0.798 | 0.816 | 0.802 | 0.900 | 0.935 |
| | | (0.823–0.830) | (0.852–0.859) | (0.792–0.803) | (0.812–0.820) | (0.797–0.806) | (0.896–0.903) | (0.930–0.939) |
| | **Transformer** | 0.851 | 0.869 | 0.810 | 0.822 | 0.814 | 0.908 | 0.941 |
| | | (0.847–0.854) | (0.866–0.872) | (0.804–0.816) | (0.819–0.826) | (0.809–0.818) | (0.904–0.911) | (0.936–0.945) |
| **SOFA** General | **LSTM** | 0.827 | 0.857 | 0.789 | 0.810 | 0.792 | 0.914 | 0.925 |
| | | (0.824–0.831) | (0.855–0.860) | (0.783–0.795) | (0.806–0.815) | (0.787–0.796) | (0.911–0.917) | (0.919–0.931) |
| | **TCN** | 0.840 | 0.862 | 0.807 | 0.817 | 0.792 | 0.915 | 0.934 |
| | | (0.837–0.843) | (0.860–0.866) | (0.801–0.813) | (0.813–0.822) | (0.787–0.796) | (0.912–0.918) | (0.929–0.939) |
| | **Transformer** | 0.844 | 0.860 | 0.807 | 0.833 | 0.809 | 0.916 | 0.941 |
| | | (0.841–0.848) | (0.857–0.863) | (0.801–0.813) | (0.815–0.823) | (0.804–0.813) | (0.912–0.918) | (0.937–0.946) |
| **SOFA** sepsis-3 | **LSTM** | 0.826 | 0.845 | 0.767 | 0.797 | 0.765 | 0.889 | 0.923 |
| | | (0.822–0.829) | (0.841–0.848) | (0.760–0.773) | (0.793–0.801) | (0.761–0.770) | (0.886–0.892) | (0.918–0.928) |
| | **TCN** | 0.823 | 0.853 | 0.780 | 0.801 | 0.770 | 0.896 | 0.919 |
| | | (0.819–0.827) | (0.850–0.856) | (0.773–0.787) | (0.797–0.805) | (0.765–0.775) | (0.893–0.900) | (0.914–0.924) |
| | **Transformer** | 0.831 | 0.852 | 0.778 | 0.802 | 0.786 | 0.896 | 0.916 |
| | | (0.828–0.835) | (0.849–0.855) | (0.772–0.784) | (0.798–0.806) | (0.781–0.790) | (0.893–0.899) | (0.911–0.920) |

[1] CHF: congestive heart failure.

[2] Diabetes: Diabetes without complications.

[3] Renal: renal disease.

[4] SLD: severe liver disease. The same acronyms are used below.

Next, we used regularization to investigate if static information provides any complementary value for the prediction. Regularization is a common technique to perform feature selection and alleviate overfitting. In the case that static information provides no complementary value, regularization would tend to push weights associated with static data to zero. We applied either L1 or L2 regularization on MLP blocks II, III, IV, V and VI (Fig 1B). We then analyzed the weights differences in these MLP layers before and after regularization and compared them with the MLP layers in the main branch (shown as the purple MLP block in Fig 1A). The results (S1 Text and S1 Fig) show that after L1/L2 regularization, weights in MLP blocks II, III, IV, V and VI are all reduced by 2 to 7 orders of magnitude, consistent with the limited added value of the static information. Both fusion and regularization results support our findings that static information is already encoded in the original time-series data and model hidden representations.

## 2.2 eICU validation

To ensure that the results obtained across two predictions, 9 models, and 18 static variables are not specific to the MIMIC-IV dataset, we performed model testing using another commonly

**Table 4. Fusion methods comparison.** Top: Average RMSE. Bottom: 95% CI.

| No fusion | Fuse I | Fuse V | Fuse VI | Fuse I, V, IV | Fuse I-VI |
|---|---|---|---|---|---|
| 1.69 (1.65–1.74) | 1.69 (1.65–1.75) | 1.69 (1.64–1.73) | 1.69 (1.64–1.73) | 1.70 (1.65–1.75) | 1.70 (1.65–1.75) |

**Table 5. eICU test results.** Average and 95% CI reported.

| | | Sex | Age | Race | CHF | Diabetes | Renal | SLD |
|---|---|---|---|---|---|---|---|---|
| **48h IHM** | **LSTM** | 0.714 | 0.767 | 0.763 | 0.735 | 0.858 | 0.822 | 0.882 |
| | | (0.709–0.718) | (0.763–0.772) | (0.756–0.768) | (0.727–0.742) | (0.849–0.867) | (0.815–0.829) | (0.866–0.898) |
| | **TCN** | 0.701 | 0.762 | 0.746 | 0.749 | 0.829 | 0.809 | 0.884 |
| | | (0.696–0.705) | (0.758–0.766) | (0.739–0.753) | (0.743–0.757) | (0.819–0.839) | (0.801–0.816) | (0.869–0.899) |
| | **Transformer** | 0.729 | 0.786 | 0.767 | 0.764 | 0.870 | 0.830 | 0.884 |
| | | (0.725–0.733) | (0.782–0.790) | (0.760–0.773) | (0.758–0.771) | (0.862–0.878) | (0.823–0.836) | (0.869–0.899) |
| **SOFA** General | **LSTM** | 0.715 | 0.766 | 0.730 | 0.757 | 0.836 | 0.825 | 0.888 |
| | | (0.711–0.720) | (0.761–0.770) | (0.723–0.736) | (0.750–0.764) | (0.826–0.846) | (0.819–0.832) | (0.872–0.903) |
| | **TCN** | 0.721 | 0.772 | 0.743 | 0.744 | 0.841 | 0.828 | 0.889 |
| | | (0.717–0.725) | (0.768–0.776) | (0.737–0.750) | (0.736–0.750) | (0.831–0.851) | (0.821–0.834) | (0.871–0.904) |
| | **Transformer** | 0.725 | 0.760 | 0.767 | 0.766 | 0.857 | 0.832 | 0.894 |
| | | (0.721–0.730) | (0.756–0.764) | (0.760–0.773) | (0.759–0.773) | (0.847–0.866) | (0.825–0.839) | (0.878–0.908) |
| **SOFA** sepsis-3 | **LSTM** | 0.654 | 0.727 | 0.709 | 0.691 | 0.789 | 0.790 | 0.849 |
| | | (0.649–0.659) | (0.722–0.731) | (0.703–0.717) | (0.685–0.698) | (0.776–0.801) | (0.784–0.796) | (0.835–0.863) |
| | **TCN** | 0.656 | 0.735 | 0.709 | 0.694 | 0.766 | 0.795 | 0.837 |
| | | (0.651–0.660) | (0.730–0.739) | (0.702–0.716) | (0.687–0.700) | (0.753–0.777) | (0.789–0.801) | (0.821–0.853) |
| | **Transformer** | 0.668 | 0.722 | 0.741 | 0.686 | 0.791 | 0.805 | 0.860 |
| | | (0.663–0.672) | (0.718–0.726) | (0.734–0.748) | (0.679–0.693) | (0.780–0.803) | (0.800–0.810) | (0.847–0.873) |

accepted EHR database: eICU. For either the SOFA/mortality prediction models in Table 2 or the static information prediction models in Table 3, they were all developed using only MIMIC-IV data. Here, we utilized the entire model pipeline and fed the eICU data into the cascaded feature extraction model and the static information prediction model without any fine tuning. The eICU model test results are in Table 5. For the complete eICU validation performance, please refer to S1 Table–S10 Table. We found that models trained on MIMIC-IV still had reasonable performance on eICU, with AUCs ranging from 0.654 to 0.894. Compared with the original MIMIC-IV AUCs, the smallest changes are -0.004 or +0.005, while the largest change is -0.172. We show age and SLD prediction confusion matrices (CM) for both MIMIC-IV test set and the entire eICU dataset in Fig 2. The rest of the cross validation CM are in S2 Fig.

Finally, eICU can also act as the training database and MIMIC-IV can serve as the testing database (S11 Table). In this case, we still observe high AUCs; eICU test set AUCs are between 0.697 and 0.878. Using the entire MIMIC-IV as the test set results in AUCs from 0.692 to 0.884, which further supports that our findings are valid across databases. Together, our results indicate that being able to predict patient static information is endemic issue with MLHC models utilizing EHR data. It applies to information beyond race, and is valid across clinical tasks, cohorts, model architectures, and databases.

## 2.3 STEER for disentanglement and utility

Since ML models can encode static variables even when blinded to those variables during training, we sought to develop a method to ameliorate the impact of sensitive attributes on the model outcomes. We used a VAE model where we partitioned the latent space into sensitive from non-sensitive latent space during training. Here we only consider sex, age and race as sensitive attributes. Under the training objective we outlined in Section 4.6, we set the non-sensitive latents $z \in \mathbb{R}^{N_z*T}$ and sensitive latents $b \in \mathbb{R}^{1*T}$. $a \in \mathbb{R}^1$ and $a$ is sex, age or race. We explore the model performance in both disentanglement and utility. Disentanglement is represented by the AUC of using sensitive latents to predict sex, age or race while utility is

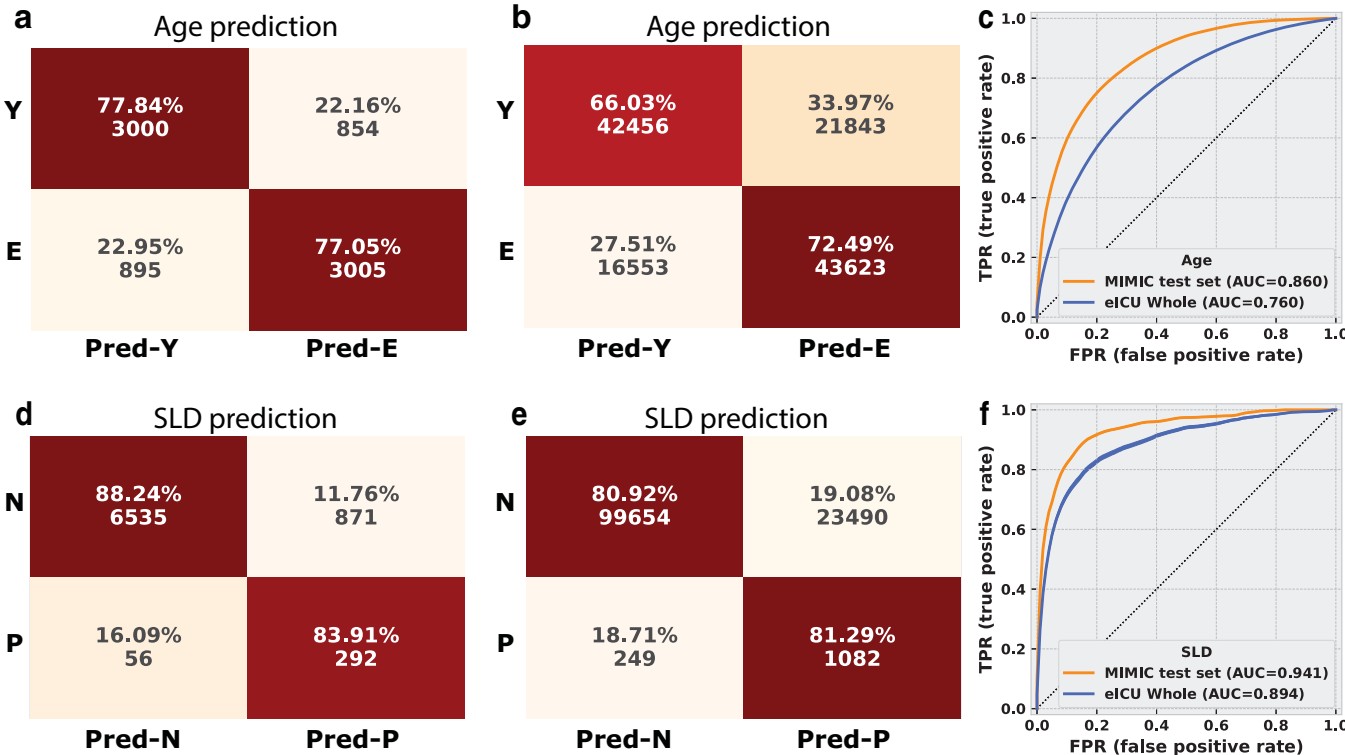

**Fig 2. Confusion matrix and AUC curves in MIMIC-IV test set and the entire eICU dataset (without fine tuning).** The hidden representations were extracted from the SOFA score prediction model from the Transformer model with General cohort. The confusion matrices on binarized age prediction for a) MIMIC-IV test set and b) the entire eICU dataset. Y: age < median age 67. E: age >= median age 67. Pred-Y: Predicted as group Y. Pred-E: Predicted as group E. c) The AUC curves for these 2 sets. Confusion matrix of the d) MIMIC-IV test set and e) the entire eICU set on SLD prediction. N: without SLD. P: with SLD. N: Predicted as group N. Pred-P: Predicted as group P. f) AUC curves.

represented by the RMSE of using non-sensitive latents to predict SOFA score. The goal is to achieve both high AUC and low RMSE. The upper bound of AUC and the lower bound of RMSE are derived from training with each goal separately using the time-series data (Original in Table 6). Training to achieve low RMSE is performed in Table 2 (RMSE: 1.69) and training to achieve high AUC is investigated in Table 3 (sex: 0.857, age: 0.876 and race: 0.833).

We incorporated multiple objectives into model training. First, we adopted the idea developed in FactorVAE [19], which optimizes both the classical VAE objective and a $\gamma$-weighted total correlation term to encourage disentanglement in the latent space. Flexibly Fair VAE [18] further suggests adding a predictiveness term which incentivizes sensitive latents to have high mutual information with the corresponding original attributes. Given the time-series nature of our EHR data, we further optimized and explored additional weighting strategies (how to weight each loss term and how to weight each record in the EHR given the various ICU stay length) in order to optimize all the components in a convenient one-stage training. Further details are in Section 4.6.

The 4 terms are the classical VAE term (also the ELBO term, controlled by $\beta$), the predictiveness term (controlled by $\alpha$), the disentanglement term (control by $\gamma$) and the clinical task prediction term (controlled by $\theta$). Here we fix $\beta$, $\gamma$, $\alpha$ ($\beta = 0.0001$, $\gamma = 0.5$, $\alpha = 0.5$) and explore how $\theta$ influences the trade-off between disentanglement and utility. $\theta$ controls the clinical task prediction term, which is the MSE loss for SOFA score prediction. We observe that when $\theta$ increases from 1 to 5, the RMSE decreases, while the AUC also decreases for all 3 cases

**Table 6. AUC and RMSE for single sensitive attribute.**

| | | θ = 1 | θ = 5 | θ = 10 | θ = 5, scale ELBO | Original |
|---|---|---|---|---|---|---|
| Sex | RMSE | 1.91 (1.87–1.96) | 1.78 (1.74–1.83) | 1.73 (1.68–1.78) | 1.75 (1.70–1.80) | 1.69 (1.65–1.74) |
| | AUC | 0.828 (0.800–0.854) | 0.808 (0.780–0.835) | 0.806 (0.780–0.830) | 0.817 (0.793–0.842) | 0.857 (0.854–0.860) |
| Age | RMSE | 1.91 (1.86–1.96) | 1.74 (1.69–1.79) | 1.87 (1.83–1.93) | 1.73 (1.68–1.78) | 1.69 (1.65–1.74) |
| | AUC | 0.837 (0.812–0.862) | 0.826 (0.801–0.852) | 0.834 (0.808–0.858) | 0.841 (0.816–0.864) | 0.876 (0.873–0.879) |
| Race | RMSE | 1.91 (1.85–1.96) | 1.75 (1.70–1.80) | 1.76 (1.71–1.82) | 1.76 (1.72–1.82) | 1.69 (1.65–1.74) |
| | AUC | 0.723 (0.669–0.777) | 0.716 (0.662–0.769) | 0.683 (0.628–0.735) | 0.755 (0.704–0.805) | 0.833 (0.828–0.838) |

(Table 6), demonstrating the trade-off between disentanglement and utility. All the RMSE and AUC metrics are also bounded by the last column in Table 6, suggesting the entangled nature of these 2 objectives. Because the EHR data has varied length T for each record, we explored normalizing the ELBO term in the loss function by $\frac{1}{T}$ and found that it improves the performance in both aspects. In predicting the sensitive labels, this approach can achieve 90.6–96.0% of the original AUC, only at the expense of increasing the RMSE by 2.4–4.1%.

The sensitive latent space can also be set to multidimensional, here we set it b $b \in \mathbb{R}^{3*T}$ and $a \in \mathbb{R}^3$ and $a$ are sex, age and race labels. We investigated the role of $a$, which controls the predictiveness term. The results show that increasing $a$ results in higher RMSE and high AUC (Table 7), as expected. Furthermore, to demonstrate the generalizability and flexibility of STEER, we applied it to a different dataset (eICU) and a different task (IHM). Notably, for the IHM task, the latent space is set to $z \in \mathbb{R}^{N_z}$ and $b \in \mathbb{R}^1$ (non-time-series case). The results are in S12 Table and S13 Table. Together, these results demonstrate the wide applicability of STEER, which permits learning representations that can be easily modified to mitigate the influence of sensitive information, enabling a wide variety of downstream tasks.

## 3. Discussion

EHRs enable the use of decision support tools and analytics to improve patient outcomes, quality of care, and operational efficiency. With the increasing adoption of EHRs and rapid development of ML algorithms, their implementation and use require careful consideration to ensure that patient privacy is maintained and fairness is achieved.

In this work, we first found that original time-series EHR as well as model hidden representations can be trained to predict a wide range of static variables with AUCs as high as 0.941. All the extracted models had no access to static information during training yet obtained high predictive performance on several static variables. More importantly, in

**Table 7. AUC and RMSE for 3 sensitive attributes.**

| | RMSE | Sex AUC | Age AUC | Race AUC |
|---|---|---|---|---|
| α = 0.5 | 1.90 (1.85–1.95) | 0.752 (0.723–0.780) | 0.764 (0.736–0.792) | 0.747 (0.699–0.795) |
| α = 2 | 2.09 (2.04–2.14) | 0.775 (0.747–0.804) | 0.806 (0.778–0.834) | 0.769 (0.719–0.818) |
| α = 5 | 2.91 (2.80–3.02) | 0.761 (0.732–0.789) | 0.804 (0.780–0.833) | 0.775 (0.726–0.821) |

order to test the findings on a larger and more diverse database and analyze if the models are learning any MIMIC-IV-specific artifacts, we directly tested all the feature extraction models and static information prediction models (all trained on MIMIC-IV) on eICU-extracted data without any fine tuning. We found that the AUC difference can be as small as -0.004 or +0.005.

Learning the static information itself does not necessarily lead to unfair consequences, but under certain circumstances it is a potential source of harm. It poses risks of re-identifying patients and perpetuating bias and discrimination against certain demographic groups [4,13,14,35]. Both MIMIC and eICU are in compliance with the Health Insurance Portability and Accountability Act (HIPAA) Safe Harbor provision, which stipulates a set of 18 identifiers (names, addresses, telephone numbers etc.) that must be removed in order for a dataset to be considered deidentified. Additionally, all users agree to protect the privacy and confidentiality of patient data and to ensure that the data is used only for legitimate research purposes before gaining data access. These measures help ensure that the data is used in an ethical and responsible manner. However, as the "fairness through awareness" framework suggests, we cannot assume sensitive information has been expunged from a dataset [36]. Our findings convincingly demonstrate this. To mitigate the potential risks of ML models learning to re-identify patients, data users should strictly follow ethical guidelines, maintain the confidentiality and privacy of any individual patient data contained in the database, take reasonable steps to prevent unauthorized access or disclosure of the data under all circumstances.

Furthermore, we also developed an algorithmic approach STEER to mitigate the potential risks of ML models learning to re-identify patients and produce unfair decisions, which aims at learning a structured latent space to disentangle the information required for clinical tasks and the information that are sensitive and should be protected. We believe STEER is widely applicable in situations where the model decisions should be blinded to certain attributes (besides sex, age and race studies here). We focused on either point or time-series prediction using EHR, but we believe it could also be adapted to work with imaging [18] or text data. Importantly, other measures are also critical in ensuring that ML tools are used in an ethical and responsible manner that benefits all patients equally. This includes strategies such as using diverse and representative training data [5], creating"Datasheets for Datasets"[37], designing models to be interpretable [38], and other approaches to achieve fair representation [26,39].

Our study has a few limitations. The conclusions from this work are established from cross validation between eICU and MIMIC-IV databases. The data in MIMIC-IV was drawn from patients who were treated at a single medical center in Boston. The eICU database was drawn from a specific subset of patients who were admitted to participating ICUs in the United States. Therefore, it may not accurately reflect the diversity of patients and healthcare practices in other regions or countries. This could limit the generalizability of the findings. Moreover, an interpretability investigation as to what drives the model to learn the static information is also out of the scope of this study. Lastly, due to the constraints of the data, we only performed binary prediction for sex, age and race, which is an over-simplification of the real-world demographics.

## 4. Methods

### 4.1. Cohort and features

We used the extraction pipeline METRE [40] to extract the ICU stay data in MIMIC-IV [41] and eICU Collaborative Research Database [42]. We used the following 2 cohorts throughout this work:

The general cohort, which is all the ICU stays in MIMIC-IV and eICU databases whose age at ICU admission was 18 years or over, and whose ICU stay was between 24h and 240h. As a result, we have 38,766 and 126,448 ICU records from these 2 databases, respectively. The second cohort, termed the Sepsis-3 cohort, termed the Sepsis-3 cohort, is a subset of the general cohort which satisfies the Sepsis-3 criteria [32], in which organ dysfunction can be identified as an acute change in total SOFA score ≥2 points and the baseline SOFA score can be assumed to be zero in patients not known to have preexisting organ dysfunction. Therefore, out of the general cohort, we selected ICU stays that had SOFA score ≥2 anytime during the ICU stay and suspected infection to be the Sepsis-3 cohort. For this cohort, we have 18,329 records for MIMIC-IV and 25,078 records for eICU. The resulting dataframes contain both static and time-series features. The list of static features includes sex, age, race, myocardial infarct, congestive heart failure, peripheral vascular disease, cerebrovascular disease, dementia, chronic pulmonary disease, rheumatic disease, peptic ulcer disease, mild liver disease, diabetes, paraplegia, renal disease, malignant cancer, severe liver disease, and metastatic solid tumor. The time-series features include lab tests (platelets, creatinine, lactate etc.), vital signs (heart rate, respiratory rate, blood pressure etc.) as well as intervention features. In total 108 time-series features were used. Despite using different model architectures and modeling cohorts to prove that our findings are generalizable, the same set of time-series features were used in all the studies conducted here. For details on the time-series features used, please refer to S14 Table.

## 4.2 Prediction tasks

### 4.2.1 48h in-hospital mortality prediction

48 hour all-cause in-hospital mortality (IHM) is a popular benchmark challenge in the community [43,44]. Our first task is to predict IHM using 48h of ICU data. Besides, we also implemented a 6h gap between the end of the observation window (48h) and the positive outcome in order to prevent the targeted task outcome leaking back into features [45] and to support clinically actionable planning [46]. Thus, we considered all stays of at least 54h in this analysis. The positive labels are documented as *hospital_expire_flag* in *mimic_core.admissions* in MIMIC-IV. For eICU, *eicu_crd.patients* record *unitdischargestatus* ('Alive' or 'Expired') explicitly.

**4.2.2 SOFA score prediction.** We used the multivariate time-series data to develop a model that iteratively gives the prediction for the patient's future SOFA score. Let $X = (X_1, X_2, X_3, \ldots, X_T) \in R^{m*T}$, denote a multivariate time series, where m is the number of variables in the time series and $X_i = (x_i^1, x_i^2, \ldots, x_i^m)$ is the measurements of the input multivariate time series at time i. The prediction $Y = (Y_{P+1}, Y_{P+2}, Y_{P+3}, \ldots, Y_{P+T})$, where $Y_{P+1}$ is the SOFA score at $P+1$ hour and is only dependent on $X_1$, $Y_{P+2}$ is dependent on both $X_1, X_2$ etc.. When making the final $Y_{P+T}$, the model is seeing the entire $X = (X_1, X_2, X_3, \ldots, X_T)$. We set the prediction horizon $P$ to be 24h in this study. The schematic of the SOFA score prediction strategy is shown in Fig 1A, where the input is the multivariate time series $X$ and the output is a predicted SOFA sequence. Each point $Y_{P+T}$ in the predicted sequence is only dependent on $X = (X_1, X_2, X_3, \ldots, X_T)$. The SOFA score ground truth was calculated based on the MIMIC-IV GitHub repository [47] for both MIMIC-IV and eICU records used in this study.

## 4.3 Models

We used the following three model architectures throughout this work to learn from the time-series data. We adopted these models mainly due to their established performance in time-series data modeling [48–50]. We define our output layer as a linear layer which takes input of

(batch size, ICU stay hour, hidden dim) and output vectors of dimension (batch size, ICU stay hour, 2) for IHM task and (batch size, ICU stay hour, 1) for SOFA score prediction. For IHM task, cross entropy loss was used during training. As for the SOFA prediction task, the prediction is also a numerical data time series, thus mean squared error (MSE) loss was computed with the ground truth SOFA score every hour.

- LSTM [51]. The number of features in the LSTM hidden states is 512 and the number of recurrent layers is 3. A Dropout layer (p = 0.2) on the outputs of each LSTM layer is used. The outputs from the last recurrent layer are passed through the output layer.

- Temporal convolutional network [52] (TCN). To guarantee that the network produces an output of the same length as the input for the SOFA score prediction, TCN uses a 1D fully-convolutional network architecture, where padding of length (kernel size minus 1) is added to keep subsequent layer output the same length as previous ones. To achieve no leakage from the future into the past, TCN uses causal convolutions, where an output at time T is convolved only with elements from time T and earlier in the previous layer, consistent with the setup we established in Section 4.2. We used 4 temporal blocks with feature map dimension 256 followed by 2 fully-connected layers (followed by RELU activation and 0.2 dropout) of dimension 128. Lastly, the output layer is used. Each temporal block is represented as TB1, TB2, TB3, TB4 in Fig 1.

- Transformer model [53]. We adopted the transformer encoder architecture to encode the multivariate time series. The number of expected features in the encoder is 256. We used 8 heads for the multi-head attention. The dimension of the feed-forward network model is 1024 and the total number of encoder layers is 2. For the decoder, we used the output layer.

## 4.4 Inferring static data from time-series

After the models were trained for their respective tasks using specific cohorts (General or Sepsis-3), learned features were extracted. For the extracted feature vectors, we used 2 fully connected layers with outputs size 128 and 2 to perform binary predictions. Prediction target includes age, sex, race, and various comorbidity factors.

Both MIMIC-IV and eICU databases are imbalanced in the race distribution (e.g., only 3.4% ICU records are Hispanic). So we focused on 2 races with the most number of records (Black/African American and White) and performed binary classification. For age, we split the entire cohort into 2 subgroups based on the median age (67) and performed binary classification. In terms of the biological sex, we didn't find unknown entries in the MIMIC-IV database but find 0.1% unknown entries in the eICU. Unknown records are excluded in the model training and testing for accurate model quantification. The remaining static information is comorbidity-related and is also binary.

## 4.5 Time-series and static data fusion

For the SOFA score prediction task using the TCN model, we studied different methods to fuse the static information and compared how they affect the performance. Give the structure of the TCN model, we performed 1) early fusion at I, 2) intermediate fusion at V, 3) late fusion at VI, 4) combined fusion at I + V + VI, 5) all-level fusion from I to VI, where the Roman Numerals I to VI are indicated in Fig 1B. At each fusion point except early fusion I, the static data is passed through a MLP model (3 layers, 256 output features with RELU activation and 0.2 dropout in between) before being replicated in the time dimension and concatenated with the time-series output in the feature dimension at each point. Under the all-level fusion

strategy, we applied L1 ($\lambda = 0.001$) or L2 ($\lambda = 0.0001$) selectively on the MLP blocks which are responsible for integrating static information. These MLP blocks are also shown as block II to VI in Fig 1B. As a control, we compared the weights differences when there is L1/L2 regularization with no regulation.

## 4.6 Learn a structured latent space

We employ the following notations:

- $X \in \mathbb{R}^{m*T}$: multivariate time series EHR.

- $a \in \mathbb{R}^1$ or $\mathbb{R}^3$: original sensitive labels.

- $b \in \mathbb{R}^{1*T}$ or $\mathbb{R}^{3*T}$: sensitive subspace of the latent space. Both $a$ and $b$ depend on whether one single attribute is targeted or all 3 attributes.

- $z \in \mathbb{R}^{N_z*T}$: non-sensitive subspace of the latent space.

  The goal of the model has 4 components:

- The first part is derived from the classical VAE[54]. VAE is trained to optimize a loss function that consists of two components: the reconstruction loss and the KL-divergence loss $D_{KL}[q(z,b|x)||p(z,b|x)]$, which penalizes the divergence between the estimated latent distribution and the unit Gaussian distribution. In practice, the objective to be maximized is following Evidence Lower Bound (ELBO), which is achieved by training an encoder and decoder. The custom-designed encoder and the decoder can be found in the online repository.

$$\mathbb{E}_{q(z,b|x)}[\log p(x|z,b)] - D_{KL}[q(z,b|x)||p(z,b)]$$

- We want high predictive such that $b$ can be used to predict $a$, mathematically, we want to maximize $\mathbb{E}_{q(z,b|x)}[\log p(a|b)]$, which is achieved by training a MLP model.

- Another desired property is the disentanglement of $b$ and $z$, which is to minimize $D_{KL}[q(z,b)||q(z)\prod_j q(b_j)]$. It is achieved by training a binary adversary to approximate the log density ratio $\log \frac{q(z,b)}{q(z)\prod_j q(b_j)}$.

- The model still preserves the ability to perform clinical task prediction. For SOFA score prediction, the goal is to minimize the mean square error. For IHM, the loss is cross entropy. We denote this term as $L_{ctp}$.

  We also found the following techniques improved the model performance: scale the ELBO loss by $\frac{1}{T}$, $T$ is the length of the time series data in each batch. Instead performing a 2-stage training and optimizing $L_{ctp}$ separately as the original paper did [18], we used a hyperparameter $\theta$ to control this term and we can optimize the above goals by maximizing the following function:

$$L = \beta/T*(\mathbb{E}_{q(z,b|x)}[\log p(x|z,b)] - D_{KL}[q(z,b|x)||p(z,b)])$$

$$+\alpha*\mathbb{E}_{q(z,b|x)}[\log p(a|b)] - \gamma*D_{KL}[q(z,b)||q(z)\prod_j q(b_j)]$$

$$-\theta*L_{ctp}.$$

### 4.7 Ethics approval statement

MIMIC-IV data was collected at Beth Israel Deaconess Medical Center (BIDMC) as part of routine clinical care is deidentified, transformed, and made available to researchers who have completed training in human research and signed a data use agreement. The Institutional Review Board at the BIDMC granted a waiver of informed consent and approved the sharing of the research resource. Study using eICU-CRD is exempt from institutional review board approval due to the retrospective design, lack of direct patient intervention, and the security schema, for which the re-identification risk was certified as meeting safe harbor standards by an independent privacy expert (Privacert, Cambridge, MA) (Health Insurance Portability and Accountability Act Certification no. 1031219–2)

## Supporting information

**S1 Text. Regularization experiments.**
(PDF)

**S1 Fig. Regularization results.**
(PDF)

**S2 Fig. eICU cross validation.**
(PDF)

**S1 Table. Predicting from original time-series variables, MIMIC-IV.**
(PDF)

**S2 Table. Feature extraction model: LSTM, IHM.**
(PDF)

**S3 Table. Feature extraction model: TCN, IHM.**
(PDF)

**S4 Table. Feature extraction model: Transformer, IHM.**
(PDF)

**S5 Table. Feature extraction model: LSTM, SOFA prediction, General cohort.**
(PDF)

**S6 Table. Feature extraction model: TCN, SOFA prediction, General cohort.**
(PDF)

**S7 Table. Feature extraction model: Transformer, SOFA prediction, General cohort.**
(PDF)

**S8 Table. Feature extraction model: LSTM, SOFA prediction, Sepsis 3 cohort.**
(PDF)

**S9 Table. Feature extraction model: TCN, SOFA prediction, Sepsis 3 cohort.**
(PDF)

**S10 Table. Feature extraction model: Transformer, SOFA prediction, Sepsis 3 cohort.**
(PDF)

**S11 Table. eICU as the training database, test on MMIC-IV.**
(PDF)

**S12 Table. STEER results on eICU database.**
(PDF)

**S13 Table. STEER results on IHM task.**
(PDF)

**S14 Table. Time-series variables in the EHR.**
(PDF)

## Acknowledgments

We thank Dr. Luca Daniel (Massachusetts Institute of Technology), Dr. Tsui-Wei Weng (University of California San Diego), Ching-Yun Ko (Massachusetts Institute of Technology) for their valuable discussions.

## Author Contributions

**Conceptualization:** Wei Liao.

**Data curation:** Wei Liao.

**Methodology:** Wei Liao.

**Resources:** Joel Voldman.

**Software:** Wei Liao.

**Supervision:** Joel Voldman.

**Validation:** Wei Liao.

**Visualization:** Wei Liao.

**Writing – original draft:** Wei Liao, Joel Voldman.

**Writing – review & editing:** Wei Liao, Joel Voldman.

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
