## [Decision Letter · Decision Letter 0]

8 Jul 2024

PDIG-D-24-00103

Learning and diSentangling patient static information from time-series Electronic hEalth Records (STEER)

PLOS Digital Health

Dear Dr. Voldman,

Thank you for submitting your manuscript to PLOS Digital Health. After careful consideration, we feel that it has merit but does not fully meet PLOS Digital Health's publication criteria as it currently stands. Therefore, we invite you to submit a revised version of the manuscript that addresses the points raised during the review process.

Please submit your revised manuscript within 60 days Sep 06 2024 11:59PM. If you will need more time than this to complete your revisions, please reply to this message or contact the journal office at digitalhealth@plos.org. Please include the following items when submitting your revised manuscript:

We look forward to receiving your revised manuscript.

Kind regards,

Hualou Liang

Academic Editor

PLOS Digital Health

Journal Requirements:

1. We ask that a manuscript source file is provided at Revision. Please upload your manuscript file as a .doc, .docx, .rtf or .tex.

Additional Editor Comments (if provided):

Reviewers' comments:

Reviewer's Responses to Questions

**Comments to the Author**

1. Does this manuscript meet PLOS Digital Health’s publication criteria? Is the manuscript technically sound, and do the data support the conclusions? The manuscript must describe methodologically and ethically rigorous research with conclusions that are appropriately drawn based on the data presented.

Reviewer #1: Yes

Reviewer #2: Yes

2. Has the statistical analysis been performed appropriately and rigorously?

Reviewer #1: Yes

Reviewer #2: Yes

3. Have the authors made all data underlying the findings in their manuscript fully available (please refer to the Data Availability Statement at the start of the manuscript PDF file)?

Reviewer #1: Yes

Reviewer #2: No

4. Is the manuscript presented in an intelligible fashion and written in standard English?

Reviewer #1: Yes

Reviewer #2: Yes

5. Review Comments to the Author

Reviewer #1: This paper provides a comprehensive and thoroughly conducted study on the potential of time-series electronic health record data to forecast patient static information. The authors have made substantial contributions to the field of machine learning, and their findings are both original and influential. The methodology is strong and reliable, the analysis is thorough and extensive, and the findings are strongly supported by the data. Therefore, I recommend this paper for acceptance.

Reviewer #2: In the submitted manuscript titled "Learning and diSentangling patient static information from time-series Electronic hEalth Records (STEER)" the author suggest a method to protect patient-sensitive information using VAE model to diSentangle patient static information from Time-series Electronic hEalth Records (STEER) by composing a latent space of two subspaces, sensitive and non-sensitive.

Comments:

- How was the General and the Sepsis-3 cohorts defined?

- SOFA RMSE are 1.45-1.48 for the General cohort and 1.69-1.72 for the Sepsis-3 cohort, which are reasonable - A reasonable measure is not defined. Are there other publications with similar results? Or Clinical guidlines?

- Add to Table 1 summary statistics of SOFA.

 - 48h mortality prediction - Do you predict anytime forward mortality, or was it limited to stays up to some certain length?

- SOFA prediction - is every hour of every patient considered as an observation in terms of training sample and test samples? Or was a random window selected to predict a single next SOFA of a patient? Were the patients/visits split to train and test without shared patients/visits?

- What were the features? This wasn't clear enough for both tasks.

- What are the 18 static features? There appears a list of only 7 in Table 3. I suggest you give the complete list in the Methods section, and don't rely on referring to the supplementary tables.

6. PLOS authors have the option to publish the peer review history of their article (what does this mean?). If published, this will include your full peer review and any attached files.

**Do you want your identity to be public for this peer review?** For information about this choice, including consent withdrawal, please see our Privacy Policy.

Reviewer #1: No

Reviewer #2: No

---

## [Decision Letter · Decision Letter 1]

11 Sep 2024

Learning and diSentangling patient static information from time-series Electronic hEalth Records (STEER)

PDIG-D-24-00103R1

Dear Dr. Voldman,

We are pleased to inform you that your manuscript 'Learning and diSentangling patient static information from time-series Electronic hEalth Records (STEER)' has been provisionally accepted for publication in PLOS Digital Health.

Best regards,

Hualou Liang

Academic Editor

PLOS Digital Health

Reviewer Comments (if any, and for reference):

Reviewer's Responses to Questions

**Comments to the Author**

1. If the authors have adequately addressed your comments raised in a previous round of review and you feel that this manuscript is now acceptable for publication, you may indicate that here to bypass the “Comments to the Author” section, enter your conflict of interest statement in the “Confidential to Editor” section, and submit your "Accept" recommendation.

Reviewer #1: All comments have been addressed

2. Does this manuscript meet PLOS Digital Health’s publication criteria? Is the manuscript technically sound, and do the data support the conclusions? The manuscript must describe methodologically and ethically rigorous research with conclusions that are appropriately drawn based on the data presented.

Reviewer #1: Yes

3. Has the statistical analysis been performed appropriately and rigorously?

Reviewer #1: Yes

4. Have the authors made all data underlying the findings in their manuscript fully available (please refer to the Data Availability Statement at the start of the manuscript PDF file)?

Reviewer #1: Yes

5. Is the manuscript presented in an intelligible fashion and written in standard English?

Reviewer #1: Yes

6. Review Comments to the Author

Reviewer #1: This manuscript is improved, the authors had revised the paper carefully. It's clear and concise now. Therefore I recommend this paper to be accepted.

7. PLOS authors have the option to publish the peer review history of their article (what does this mean?). If published, this will include your full peer review and any attached files.

**Do you want your identity to be public for this peer review?** For information about this choice, including consent withdrawal, please see our Privacy Policy.

Reviewer #1: **Yes: **Endah Kristiani
